# Patient preferences for Remote cochlear implant management: A discrete choice experiment

Catherine Sucher[1,2]*, Richard Norman[3,4], Emma Chaffey[1,2], Rebecca Bennett[1,2,5,6], Melanie Ferguson[4,6]

1 Ear Science Institute Australia, Nedlands, Western Australia, Australia, 2 University of Western Australia, Crawley, Western Australia, Australia, 3 School of Population Health, Curtin University, Bentley, Western Australia, Australia, 4 Curtain Enable Institute, Curtin University, Bentley, Western Australia, Australia, 5 National Acoustic Laboratories, Sydney, New South Wales, Australia, 6 Curtin School of Allied Health, Curtin University, Bentley, Western Australia, Australia

* catherine.sucher@uwa.edu.au

## Abstract

### Background

The opportunity to assess cochlear implant outcomes remotely provides the potential to streamline delivery of care for cochlear implant users. However, the conditions required for its implementation into clinic systems must be fully understood to ensure success and sustainability. The objectives of this study were to (i) use a discrete choice experiment quantify the preferences of cochlear implant users when considering use of Cochlear Remote Check™, a remote assessment service, and (ii) explore the perceptions, insights and attitudes of CI users that may influence utilisation of a remote service.

### Design

A discrete choice experiment was administered to Australian adult cochlear implant users via an online survey. Participants chose between pairs of hypothetical clinical service options for three different clinical scenarios (acute care, troubleshooting and long-term review). Participants answered a series of questions focusing on how and when remote services should be discussed and offered within their hearing journey.

### Results

A total of 124 adult cochlear implant users completed the survey. Conditional logit analysis revealed the strongest participant preference was clinician continuity for assessment review, followed by low service costs. They preferred to receive assessment results within one week of completion, but not by videoconference/call in the acute care scenario. Only 12% of participants preferred in-clinic visits for all scenarios. Notably, 100% of participants felt that cochlear implant users should be made aware of remote service opportunities available to them.

**Data availability statement:** The dataset includes potentially sensitive and identifying patient information. The Human Research Ethics Committee at The University of Western Australia does not normally allow public dissemination of data relating to human participants. Requests for access to de-identified data should be directed to Cochlear Implant Research Manager, Ear Science Institute Australia for researchers who meet the criteria for access to confidential data. Address: 2/1 Salvado Rd, Subiaco WA 6008 Ph: 1300 847 395 Website: earscience.org.au Email: enquires@earscience.org.au

**Funding:** This research was supported by an Investigator Initiated Research Grant from Cochlear Ltd (Grant no. 2021/ET001071), for whom author CS was the principal investigator. Cochlear Ltd had no role in the design, or conduct of the study. Members of the Cochlear ANZ clinical research team provided support in participant recruitment by forwarding email invitations to participate to individuals who had previously agreed to be contacted by Cochlear Ltd for research. Cochlear Ltd had no role in the collection, analysis or interpretation of the data. Preparation of the manuscript was supported by an independent medical writer, supported by Cochlear Ltd Australia. The initial IIR grant funding included publication costs for any articles which arose from the study.

**Competing interests:** Author CS [Recently completed a REDI Fellowship in partnership with Cochlear Ltd and Ear Science Institute Australia] Author RN [Declares no conflict of interest] Author EC The organisation funding EC's research position (Ear Science Institute Australia) as part of a PhD scholarship has received additional funding for research projects from Cochlear Ltd not related to this work. EC has not personally received any financial support from this funding. Author RB . The organisation employing RB (National Acoustic Laboratories) has received funding for research projects from Cochlear Ltd not related to this work. RB has not personally received any financial support from this funding. Author MF. The organisation employing MF (Curtin University) has received funding for two research projects from Cochlear Ltd not related to this work. MF has not personally received any financial support from this funding. The authors note that there are no patents, or products in development associated with this research. It is noted that Remote Check is a product of

## Conclusion

Study participants placed high importance on clinician continuity, but preferences for timing and delivery of results were less pronounced. This information can help to inform customisation of remote services by individual clinics. Costs and payment infrastructure for providing remote care require careful consideration. Whilst there is an appetite for use of Remote Check™ alongside clinic visits, it is not suitable for, nor preferred by, all cochlear implant users.

## Introduction

Cochlear implants (CIs) are effective devices to aid sound and speech perception for many people with severe or profound sensorineural hearing loss who experience limited benefits from hearing aids [1,2]. The World Report on Hearing [3] estimated that around 163.5 million people globally suffer from moderately-severe or poorer hearing, most of whom may be eligible for a CI. Although only 5% to 10% of eligible adults in developed countries currently access CIs [1], the number of CI users is increasing. Market analysis reports forecast an annual growth rate of 6–9% for the global CI market, driven by an aging population, increased efforts to raise awareness about CIs, and expansion of both unilateral and bilateral candidacy criteria [4–7].

Successful hearing rehabilitation with a CI is a complex, multi-stage process [8], with lifetime follow-up care required [9]. Guidelines recommend frequent appointments in the first year following activation, with routine long-term follow up every 6 or 12 months thereafter for most adults [9–11]. Although routine scheduled assessments provide opportunities for audiologists to detect deterioration in hearing that may have gone unnoticed by the CI user, they can also lead to CI users with stable hearing attending clinic-initiated appointments that provide little benefit to the user [12,13]. Moreover, the need to provide long-term follow-up care for CI users whilst also providing services to the expanding number of new CI users places an increasing, cumulative workload on implant clinics [1,7].

### Remote technologies in audiology

Remote technologies have the potential to enhance service delivery for adults with hearing loss [14]. Carrying out long-term follow-up CI assessments remotely may help reduce the burden of routine clinic visits on clinic resources and help reduce travel time/costs, time off work and disruption to family life associated with clinic visits for CI users [13–16]. Several published studies have examined the feasibility of remote care options for CI users, and have highlighted the desirability of remote care [14–22] both using manufacturer-based and stand-alone remote care options. Furthermore, since completion of this study, all three major CI manufacturers, Cochlear Ltd, Med-El and Advanced Bionics have released remote care options.

Despite the potential advantages, several barriers need to be addressed when considering integration of remote technologies into audiology clinical practice [14,19]. A systematic review by Kruse et al. [23] identified limited technical literacy, which

Cochlear Ltd, however the none of the authors of this paper, nor the research associated with this paper, are associated with is production, development or marketing. The affiliations noted in this statement to not alter the authors adherence to PLOS ONE policies on sharing data and materials.

may encompass not only a lack of familiarity with computers and smartphones (i.e., digital literacy), but also lack of trust and concerns over privacy and security, lack of desire (an attitude of "as it was not needed before, why bother to learn it now?"), and out-of-pocket costs as common barriers to using telehealth for older adults (aged ≥50 years) [23]. Audiologists may therefore be reluctant to offer remote services to older clients due to a perception of poor technical literacy and low confidence to use remote devices and services [14,24]. Moreover, implementation of new clinical workflows required for remote care programs can be challenging for audiology clinics [18,21,25]. For example, Nassiri et al. [21] identified overcoming the inertia of existing clinical workflow and the need for a designated coordinator with dedicated time to ensure effective execution of the process as potential obstacles. The clinical and administrative costs, including access to funding and funding infrastructure for services, associated with the implementation and provision of remote services must also be considered [14].

## Optimising implementation design

Given the potential advantages and pitfalls of remote service options, it is important to ensure that a well-designed implementation pathway is used to effectively employ such services within the clinical environment. The APEASE (Acceptability, Practicability, Effectiveness, Affordability, Side-effects or safety, and Equity) criteria can be used to guide development of implementation plans [26,27], and have been used previously in audiology for the implementation of digital technologies [24]. Surveys, focus groups and discrete choice experiments (DCE) can help determine the thoughts and preferences of stakeholders, and may be employed to assess the acceptability and practicability aspects of the APEASE criteria. DCEs such as the one used in this study present respondents with a series of hypothetical choices, typically between competing options. Through regression analysis, the relative importance of different dimensions of each option can be estimated, which can flow directly into policy making.

As part of a plan to investigate implementation of a remote CI care service at a large clinic in Western Australia, the current study conducted a DCE to assess CI user preferences for service options with Remote Check™ (Cochlear Ltd, Sydney), the only commercially-available clinical tool for remote CI assessment existing at the time.

## Remote Check

Remote Check™ is designed for use with CI users fitted with Cochlear Ltd (Sydney, Australia) Nucleus 7, Kanso2, or more recent sound processors. Remote Check™ allows CI users to conduct remote self-testing of hearing function, with asynchronous clinical review of the results to support patient-management decisions [15,17,20,28]. A detailed description of the Remote Check test battery is provided elsewhere [15,17,20]. Briefly, it takes about 20–40 minutes to complete and includes: photographs of implant site; self-testing of hearing outcomes with the digit triplet test (DTT), and aided threshold test (ATT) using wireless streaming via Bluetooth; 12-item

Speech, Spatial and Qualities of Hearing Scale (SSQ12) questionnaire; automated impedance test; and collection of usage data and sound processor diagnostics. Whilst some of the assessments used in Remote Check (e.g., the DTT and ATT tests) may not be exactly the same as those used in standard clinical practice, research has shown that the Remote Check application is highly successful (99%) in identifying all issues recognized by a clinician during an in-person session. [17].

Clinical evaluations of Remote Check[TM] report that it allows comprehensive, easy and reliable self-testing of hearing function by the CI user or their carer in their own home, and asynchronous access to test results by clinicians to assist in monitoring and triaging individuals for appropriate management [15]. When compared with in-clinic appointments, the test battery identified 94% of clinical issues, and identified 99% of CI users requiring clinical action [17,20]. The Remote Check[TM] outcomes of ATT or DTT were not affected by time of testing, motivation or task-related fatigue [28].

## Objectives

Although DCE studies are being increasingly used to measure preferences in healthcare and health economics [29–33], there have been few such studies in the field of audiology to date [34–37], and none examining remote services or CIs.

The objectives of this study were to (i) use a DCE to quantify the preferences of CI users when considering a Remote Check[TM] service, and (ii) to explore the perceptions, insights and attitudes of CI users that may influence utilisation of a remote service.

## Methods

### Survey participants

Potential participants were recruited opportunistically through contacts at Cochlear Australia, the Ear Sciences Institute Australia (Perth), and the Cochlear Care Centre (East Melbourne) with invitations to participate emailed to 382 adult CI users identified as meeting inclusion criteria; adults aged ≥ 18 years, using Cochlear Nucleus implants (CI632, CI622, CI612, CI532, CI522, CI512, CI24M, CI24R and CI24RE) implant/s and sound processor/s (Nucleus 7 and Kanso 2) compatible with Remote Check[TM] (Nucleus 7 or 8) and who had used at least one CI for 6 months or more. All individuals emailed had agreed to be contacted for research communications and had provided a contact email. Participant recruitment was undertaken between 17/07/2022 and 16/11/2022.

Informed consent was obtained online. Participants were provided with the Participant Information Form, and contact information for the research team (email and phone number) if they wished to ask additional questions prior to the completion of the survey. Participants were asked to indicate that they understood the information provided and consented to participate in the online survey. Participants could only proceed with the survey if they consented to participate. If they did not consent, they were unable access the survey. Participant responses to the consent statements were recorded as part of the Qualtrics survey responses. Ethics approval was provided by the UWA HREC (Project no. 2021/ET001071).

### DCE design

A DCE is a quantitative stated preference method that can be used to assess the strength of participants' preference for different aspects ('attributes') of a hypothetical service, thereby helping to design a service in which the needs of users have been considered [33]. Key features of a DCE are highlighted in **Fig 1**.

For this study, DCE attributes, levels and scenarios were identified through literature review and online focus groups. A review of the literature identified within-clinic and remote methods for providing CI user management; determinants of success with each methodology were assessed. Online focus groups covering remote CI care were conducted with adult CI users (one group with Remote Check experience [n=5] and one group with no Remote Check experience [n=5]). For the current study, a rapid inductive thematic analysis was completed to identify key issues of importance to CI users (S1 Fig Supplementary Appendix) [14].

Based on the literature review, focus groups and clinical experience with Remote Check, a list of scenarios (clinical situations in which Remote Check might be used) and attributes was determined and screened for appropriateness by members of the research team. The final list of scenarios, attributes and levels (Table 1) was developed using an iterative approach over six meetings of the research team. An example DCE question based on these scenarios, attributes and levels is shown in Fig 2.

| A hypothetical service is characterised by multiple variables ('attributes'), each of which has various 'levels' | Attributes and levels are combined to form hypothetical service packages ('options') | Participants choose between multiple options for a specified scenario | DCE data are analysed using statistical modelling |
|---|---|---|---|
| Attributes: Key features of a service<br>Levels: Values or outcomes associated with an attribute | Options are designed to capture key parameters that might influence choice | Reflects real-world situations where people decide which features are 'must have' essentials and which can be tolerated, if necessary, as part of an overall package | Preferences are compared with a reference level to generate a positive (preferred) or negative (less preferred) coefficients |

DCE: Discrete choice experiment

**Fig 1. Key features of a DCE [ 29–32].**

**Table 1. Final DCE clinical scenarios, attributes and levels utilised in choice pair options for a remote service provision offered via Remote Check. An example of one of the DCE questions (scenario and option pair) is shown in Fig 2.**

| Type of service | Scenario description | | | |
|---|---|---|---|---|
| Troubleshooting | Imagine you are having a problem with your cochlear implant and it is suggested that you could complete a Remote Check test to test your cochlear implant function at home instead of waiting for a clinical appointment. | | | |
| Long-term review appointment | Imagine you are due for your yearly cochlear implant review and it is suggested that you complete a Remote Check test of your cochlear implant function at home. An optional, short "in-clinic" appointment is available after your Remote Check if it is required. | | | |
| Acute care | Imagine you have recently got your cochlear implant. You aren't due for another clinical appointment for a while, or you can't make it to your next appointment, but you would like reassurance that things are progressing between appointments. You are offered a Remote Check test. | | | |
| **Attributes** | **Levels** | | | |
| Who reviews Remote Check test? | Trained administration staff, who will refer to an audiologist if the test indicates a problem | Any trained audiologist; I don't mind whether it's my regular audiologist or not | My regular audiologist | |
| What information to provide regarding outcome of Remote Check? | A response only if the test indicates there was a problem | A response to indicate whether the overall results were good or bad and what to do next | A detailed comparison of my hearing and speech tests with my previous results | |
| Timing of feedback | The next day | Within a week | Within two weeks | |
| How outcome information is received | A notification through the Nucleus Smart App | A report emailed to me | A videoconference or phone call from my audiologist | A face-to-face meeting with my audiologist |
| Cost to access[1] services | $10 each time my Remote Check test results are reviewed by the clinic | $30 each time my Remote Check test results are reviewed by the clinic | $40 annual fee for an unlimited number of checks | $120 annual fee for an unlimited number of checks |

[1]Cost options were determined based on a previously published study [19]

**Scenario:** Imagine you are due for your yearly cochlear implant review and it is suggested that you complete a Remote Check test of your cochlear implant function at home. An optional, short 'in-clinic' appointment is available after your Remote Check if it is required. You are offered two options for your Remote Check review, described below. If you had to choose between them, which would you pick?

| | OPTION A | OPTION B |
|---|---|---|
| **After I submit my Remote Check my test will be reviewed by…** | any trained audiologist; I don't mind whether it is my regular audiologist or not | my regular implant audiologist |
| **After I submit my Remote Check test, I will be sent…** | a response to indicate whether the overall results were good or bad and what to do next | a response only if the test indicates there was a problem |
| **Timing wise, I will get my Remote Check test results…** | the next day | within a week |
| **Remote Check test results will be provided to me by…** | a notification through the Nucleus Smart App | a report emailed to me |
| **Remote Check will cost me…** | $40 annual fee for an unlimited number of checks | $10 each time my Remote Check test results are reviewed by the clinic |
| | ◯ OPTION A | ◯ OPTION B |

**Fig 2. An example of one of the DCE scenarios and option pairs presented to the participants.** A full list of all the scenarios, attributes and levels is shown in Table 1.

## Survey methodology

The DCE design was developed using Ngene 1.2.1 software, and consisted of 360 choice pairs of hypothetical service options, split into 30 blocks of 12 choice pairs (questions) each. Presentation of blocks to survey participants was randomised within Qualtrics software. In addition to a detailed description of the tasks associated with Remote Check™ and the DCE questions, the survey included participant demographics, hearing device use, knowledge of and experience with Remote Check™, travel time to attend in-clinic appointments, and ease of making time to attend CI appointments. Participants were also asked to indicate under what circumstances they would consider using Remote Check™, and the ideal timing for discussion of Remote Check™ as an option for care (see Survey questionnaire, Supplementary Appendix).

A pilot survey was created and reviewed for wording, appropriateness and clarity by a group consisting of two CI clinicians, two front-desk staff working with CI groups, and two CI users. Following feedback, revisions were made to the survey as required to improve usability. The final online survey was developed using Qualtrics software and distributed via email in 2022 (see supplementary data for survey example).

Demographic factors were analysed to determine which factors were associated with individuals' willingness to consider remote check reviews of various formats in the future. A combination of t-tests (where data were normally distributed, ranked sign tests (where categorical data was recorded and could be ranked in order of strength) and $Chi^2$ tests for categorical data that could not be ranked, were used for analysis.

The DCE data were analysed using the conditional (fixed-effects) logit model (STATA 18 software) to identify groups of responses, and descriptive statistics in relation to demographic questions. The conditional logit model is the most common approach to analysing DCE data in health, and is standard in situations where the key area of interest is the mean preference of respondents [29].

## Results

### Demographics

Of the 382 invitations sent, 246 surveys were started and 124 participants completed >95% of the survey (120 fully completed, 4 completed all DCE questions but some demographic data missing), giving an overall response rate to the initial invite of 32% and a completion rate for initiated surveys of 50%. Power calculations for DCE surveys are notoriously difficult [38], but author RN, an expert in DCE methodology, indicated that participant numbers of ≥100 should ensure the effect size precision in analysis of the data was maintained. Each of the 30 DCE blocks was completed by 2–7 participants.

Demographics of participants with >95% completed DCE questionnaires are shown in **Table 2**. The median age of participants was 64 years (range 26–89, mean(SD) =63.7±13.1). Ninety-six percent of participants had been using their first implanted CI for a year or more.. Participants were asked to indicate their agreement with the statement "In general I am easily able to make time to attend my CI appointments." Most (80%) agreed or strongly agreed with the statement Only 12 participants (10%) disagreed or strongly disagreed with the statement. Less than half (40%) of the participants had heard of Remote Check™ before participating in the study.

Participants were asked to select one or more options relating to the type of remote check review they may consider using in the future. These included, 1) trouble-shooting appointment, 2) additional monitoring during the first 6 months post switch-on, 3) completion of the speech and hearing component of their annual review, 4) a standalone annual review without coming into clinic, 5) only coming into the clinic, no remote check. Only 15 (12%) of participants would not consider any form of remote check review appointment in the future. Statistical analysis of factors which could influence individuals' consideration of remote check reviews is shown in S1 Table. Older participants were significantly less likely to accept remote check appointments for trouble-shooting ($t$=-2.6, df=121, p=0.012), additional monitoring in the first 6 months post-CI (U(46,77) = 1190.5. p=0.002), completion of speech and hearing assessment ($t$=-3.2, df=121. p=0.002), and more likely to only want in-clinic appointments only ($t$=-2.7, df=122. p=0.007). Participants with lower annual household incomes were significantly more likely to reject troubleshooting appointments (U(81,42) = 676.5. p=0.035), and participants less able to make time for in-clinic appointments were significantly more likely to accept long-term monitoring remote check appointments (U(81,42) = 1258.0. p=0.042. No other demographic factors were significantly related to consideration of future remote checks.

### DCE outcomes

The DCE analysis (pooled across all three scenarios) is shown in **Table 3**. The levels within each attribute were compared with a reference level and a resulting co-efficient calculated. Statistical significance was p <0.05.

Compared with a reference level of 'trained administration staff', participants expressed preferences for the Remote Check™ data to be reviewed either by any audiologist or by their own audiologist, with the strongest preference being for their own audiologist (coefficient 0.68, p<0.001). Preferences were less pronounced for the type of information included in a Remote Check report, with a slight preference for a 'detailed comparison of my hearing and speech tests with my previous results' over the reference level of 'response only if test indicates a problem' (coefficient 0.2, p=0.037). Regarding timing for receiving results, there was no significant difference between 'within one week' and the reference level of 'next day' (coefficient -0.062, p=0.452), suggesting that a time frame of up to one week was acceptable, but 'within 2 weeks' was less acceptable (coefficient -0.30, p=0.001). For notification of results, there was no significant difference between the reference level of 'notification through the Nucleus Smart app' and the levels 'a report emailed to me' or 'videoconference or phone call from my audiologist'. However, a face-to-face appointment with an audiologist was the least attractive level compared to a notification (coefficient -0.26, p=0.022). As expected, participants preferred lower costs, with the reference level of AU$10 per review being preferred over AU$30 per review and an annual fee of AU$40 or AU$120. The pricing

**Table 2. Demographics for participants who completed all DCE choice options (i.e., >95% completion of the survey, n=124).**

| | |
|---|---|
| **Age:** | |
| Median (range), years | 64 (26–89) |
| <40 years, n (%) | 3 (2%) |
| 40–49 years, n (%) | 20 (15%) |
| 50–59 years, n (%) | 21 (17%) |
| 60–69 years, n (%) | 37 (23%) |
| 70–79 years, n (%) | 29 (24%) |
| ≥80 years, n (%) | 14 (11%) |
| **Gender: n (%)** | |
| Male | 76 (61%) |
| Female | 48 (39%) |
| Other/Prefer not to say | 0 |
| **Experience with Remote Check: n (%)** | |
| I had not heard about Remote Check before this survey | 75 (60%) |
| I had heard about Remote Check but I have not used it | 16 (13%) |
| I have used Remote Check before | 33 (27%) |
| **Hearing device use: n (%)** | |
| CI in both ears | 40 (32%) |
| CI one ear, HA other ear (poor hearing with HA in that ear) | 23 (19%) |
| CI one ear, HA other ear (good hearing with HA in that ear) | 24 (19%) |
| CI one ear, other ear no HA (poor hearing in that ear) | 17 (14%) |
| CI one ear, other ear no HA (good hearing in that ear) | 19 (15%) |
| No response | 1 (1%) |
| **Duration of CI use: first implanted ear (n)** | |
| Less than 6 months | 4 (3%) |
| 1 year to <2 years | 11 (9%) |
| 2 years to <5 years | 42 (34%) |
| 5 years to <10 years | 35 (28%) |
| 10 years to <20 years | 29 (23%) |
| ≥20 years | 2 (2%) |
| No response | 1 (1%) |
| **Travel time to clinic: median (range), hours** | 1.0 (0.1–40.0) |
| **I can easily find time to attend CI appointments** | |
| Strongly disagree | 4 (3%) |
| Disagree | 8 (6%) |
| Neither agree nor disagree | 12 (10%) |
| Agree | 48 (39%) |
| Strongly Agree | 48 (39%) |
| No response | 4 (3%) |
| **Out-of-pocket costs for clinic appointment[1]: median (range), AU$** | $47.50 ($0–700) |

AU$: Australian dollar; CI: Cochlear implant; DCE: discrete choice experiment; HA: hearing aid.

[1]Additional non-refundable payments incurred for travel and/or implants services by patients excluding government and/or private health insurance reimbursement.

**Table 3. DCE results – pooled across all scenarios (conditional fixed-effects logistic regression) (n=124).**

| Attributes | Levels | Coefficient | p-value | 95% CI |
|---|---|---|---|---|
| Who reviews Remote Check test | Trained administration staff | Reference level | | |
| | Any trained audiologist | 0.325 | 0.001 | 0.141; 0.509 |
| | My regular implant audiologist | 0.683 | <0.001 | 0.469; 0.898 |
| Information provided regarding outcome | Response only if test indicates a problem | Reference level | | |
| | Response indicating good/bad result and next steps | 0.062 | 0.461 | −0.103; 0.228 |
| | Detailed comparison with previous results | 0.199 | 0.037 | 0.012; 0.387 |
| Timing of feedback | The next day | Reference level | | |
| | Within a week | −0.062 | 0.452 | −0.223; 0.099 |
| | Within two weeks | −0.300 | 0.001 | −0.473; −0.128 |
| How outcome information is received | Nucleus Smart App notification | Reference level | | |
| | Emailed report | −0.020 | 0.845 | −0.216; 0.177 |
| | Videoconference/phone call with my audiologist | −0.195 | 0.053 | −0.392; 0.002 |
| | Face-to-face meeting with my audiologist | −0.256 | 0.022 | −0.476; −0.036 |
| Cost to access services | AU$10 per review | Reference level | | |
| | AU$30 per review | −0.490 | <0.001 | −0.686; −0.294 |
| | AU$40 annual fee (unlimited checks) | −0.242 | 0.035 | −0.467; −0.017 |
| | AU$120 annual fee (unlimited checks) | −1.042 | <0.001 | −1.317; −0.768 |

CI: Confidence interval; DCE: Discrete choice experiment

structure of AU$40 annual fee with unlimited checks (coefficient -0.24, p=0.035) seemed preferrable to being charged AU$30 per review (coefficient -0.49, p <0.001).

An analysis of DCE results per scenario demonstrated preferences similar to the pooled analysis (S1 Table, Supplementary Appendix). However, although there was a strong preference for CI user's own audiologist to review Remote Check™ data for troubleshooting and acute care, 'any trained audiologist' and 'my regular audiologist' were both considered acceptable for a long-term review appointment (coefficient 0.707, p<0.001 and coefficient 0.801, p<0.001, respectively). The preference for receiving results from Remote Check™ within a one-week timeframe was largely driven by the troubleshooting scenario, as there were no significant differences in preferences between 'within one day', 'within one week' and 'within two weeks' for the long-term review or acute care scenarios. The slight preference for receiving a detailed comparison of Remote Check™ results with previous results was significant for the long-term review scenario (coefficient 0.344, p=0.027), but not for the troubleshooting or acute care scenarios. The only significant outcome for notification was a preference against results being provided via videoconference/call in the acute care scenario (coefficient -0.459, p=0.004). There was no significant difference between a cost of AU$40 per *annum* and the reference level of AU$10 per test in any of the individual scenario analyses.

### When to discuss and proposed circumstances for use of Remote Check

All survey participants indicated that audiologists should discuss the option of remote care options with CI users at some point in their CI journey (Fig 3). The most commonly proposed timings for such a discussion were 'When discussing which device to get prior to having the implant surgery' or 'After the implant fitting is stabilised and the CI user is comfortable with how the implant works (3–6 months after implantation)'. Over 40% of participants indicated that an awareness of the ability to access their cochlear implant care remotely post-operatively, may have influenced their choice of CI brand..

At least 65% of participants felt they would be comfortable using Remote Check™ for troubleshooting, long term monitoring and/or to complete the speech and hearing test components of an appointment, with only 15 participants (12%) indicating they would not use Remote Check™ in any of the proposed clinical situations (Fig 4).

**Based on my knowledge of Remote Check, and my own cochlear implant experience, I think the best time to be told about Remote Check is…**

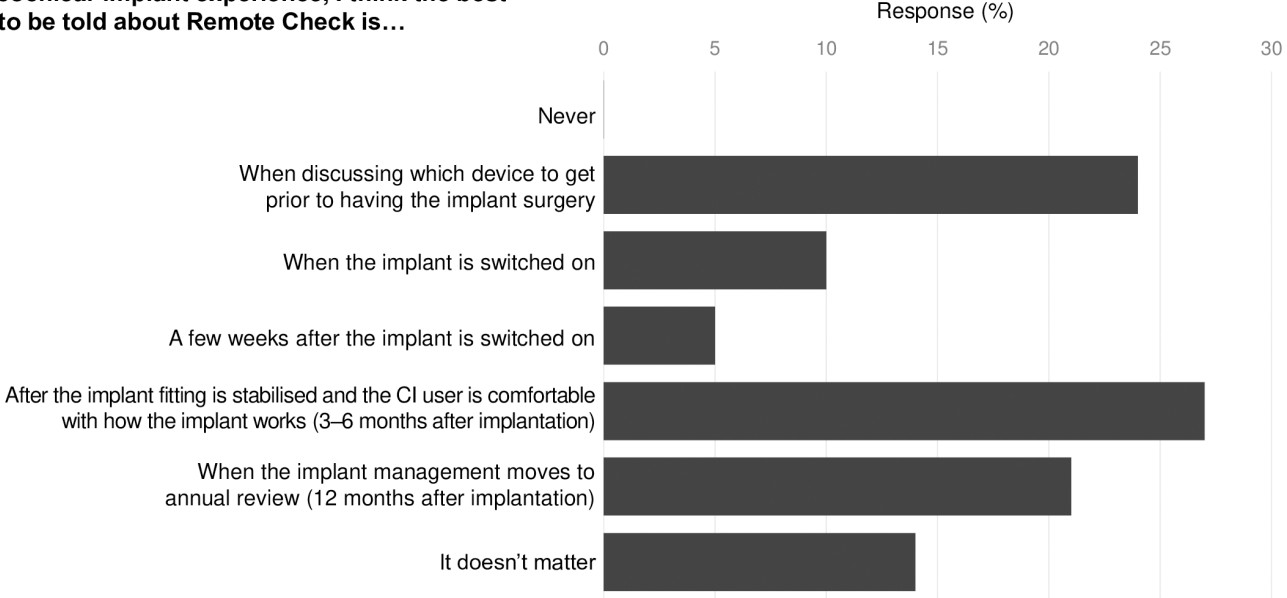

**Fig 3. Preferred timing for discussion about Remote Check.**

**Based on my understanding of how Remote Check works, I would be comfortable using Remote Check in the following situations... (tick all that apply)**

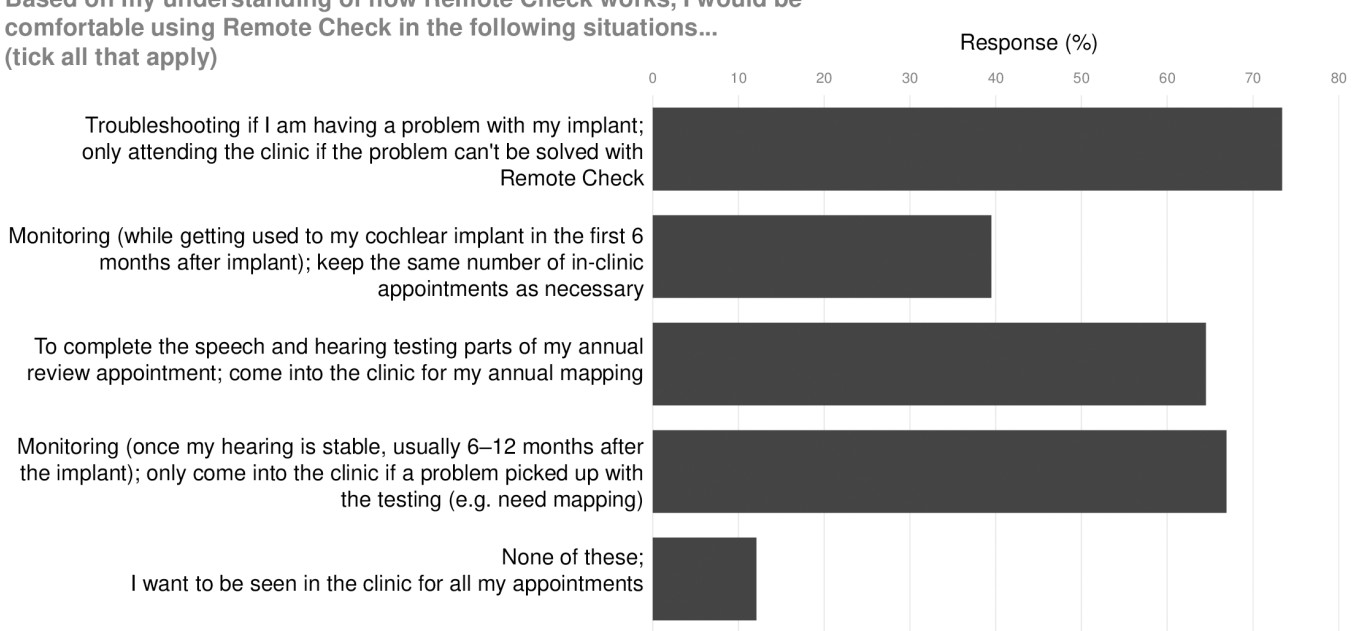

**Fig 4. Situations in which survey participants would consider using Remote Check™.**

## Discussion

This study used a DCE to explore the preferences of CI users for a Remote Check™ service model, and identified situations in which participants would consider using a Remote Check™ option. The DCE approach allowed a number of potential attributes for a Remote Check™ service to be examined using hypothetical scenarios. Drawing on the stated preferences of CI users can help audiology clinics consider what service options to offer their clients. The service model may be adapted to suit different clinics by examining individual attributes of the DCE. For a general model applicable to a broad range of circumstances, pooled data from the three different scenarios can provide an overview for each attribute. However, if a clinic wished to implement Remote Check™ for a specific circumstance (e.g., only during annual reviews), the disaggregated data for individual scenarios may be helpful.

The strongest preference demonstrated by participants in this study was for the Remote Check™ data to be assessed by a CI user's regular implant audiologist, although any trained audiologist was also acceptable, particularly for assessment of long-term review test results. This finding aligns strongly with the perceived importance of the relationship between the CI user and their audiologist that was identified in the initial focus groups. To provide a service that is preferred by clients, this study suggests that clinics should avoid using trained administration staff to review Remote Check™ assessments. However, the DCE analysis coefficient of 0.6 indicates that survey participants did not exclusively select options that included their regular implant audiologist (which would have generated a coefficient of 1.0). Thus, CI users may be willing to trade review by their regular implant audiologist with other attributes of a Remote Check™ service.

It is possible that the preference for audiologist review of Remote Check™ was driven by the overriding experience of the majority of respondents who currently receive their implant care via clinic appointments with audiologists, and have little or no experience with remote services. Moreover, survey participants may not have fully understood how Remote Check™ data can be used to triage CI users and identify those requiring follow up with their regular audiologist. Remote (telephone-based) digital triage is commonly used in other situations such as emergency care, where user satisfaction with such clinician and non-clinician led digital triage is generally high [39]. Preferences of CI users for a Remote Check™ service may evolve with increased exposure to remote care and awareness of how it can work within the service as a whole, however one cannot negate the strong relationship and sense of trust that is built between audiologists and their clients when navigating the complexity of a CI journey.

Expected timing and type of feedback of Remote Check™ results is an important consideration for allocation of clinic resources. Provided results were available within one week, participants did not consider immediate (next day) feedback to be important, but longer delays (within 2 weeks) were less favoured. This was particularly so for the troubleshooting scenario, when CI users may desire rapid resolution of any issues. Overall, participants showed a slight preference for more detailed reporting of results, primarily driven by responses to the review scenario, in which a detailed comparison with previous results may provide a useful assessment of any long-term changes. Less detailed reports were considered sufficient for troubleshooting (when follow-up action by the clinic is likely) or acute care (when reassurance is sought between in-person appointments) scenarios. Again, this may be a response to the current lived experience of CI users in our sample in relation to their audiological care. CI users are used to receiving a particular set of information, therefore believe the same information should be available from remote services; preferences may change with increased exposure to Remote Check™. It is clear, however, that in order to build trust and acceptance of Remote Check™ within the CI user body, careful consideration of how and when feedback is provided, in a clinically efficient manner, is of importance. This may involve strategies such as dedicated times within the clinical calendar for Remote Check™ review, and/or the development of templates for provision of rapid and efficient feedback, including test results, to users. Poor integration with electronic medical records has been found to be a clinician-based barrier to use of health apps in other medical fields (e.g., cardiovascular care) [40]. This is also an important consideration in how a remote digital review option, such as Remote Check™, might be integrated into current electronic clinical records results to ensure an integrated, hybrid service provision [14].

Participants preferred not to attend an in-clinic appointment to receive Remote Check™ results. This seems reasonable, as having to attend the clinic in person to obtain results negates some of the benefits of carrying out the tests remotely. However, the survey was completed at a time when the impact of COVID-19 was still apparent. It is possible that the preference to avoid in-clinic appointments was a response to the mandated or preferred avoidance of public spaces during this period [19,41–43]. Notification of Remote Check results via app, email or videoconference was usually considered acceptable by survey participants. One exception was the acute care scenario, for which a videoconference/call was not preferred. This may be due to difficulty or lack of confidence in using a telephone (or video call) that CI users often feel at a time when they are just starting to get used to their implant [44].

Costs for providing remote care service options are an important consideration. Although previous studies with Remote Check™ provide useful guidance on practical elements of remote assessments [15,17,20,28], most of these studies were carried out with public or industry funding and did not examine client preferences or implementation issues, such as integration into clinical practice and cost to the clinic. Asynchronous review of audiology data generated by Remote Check™ might be expected to free up clinic appointment time and allow clinics to see more clients, providing a cost benefit. But in Australia at least, government (Medicare) funding for CI care was based on either in-person or synchronous teleconference/phone appointments. Asynchronous review and reporting of results via an app or email were not government-funded in Australia for CI services at the time of the survey. Australian government (Medicare) rebates of between AUD222 (1 implant) to AUD409 (2 implants) could be claimed for in-clinic annual review appointments consisting of aided threshold and speech testing and implant review. Additional fees may also have been charged for some private CI services. One must also consider the client cost of attending an in-person appointment. In Australian private health insurance does not cover CI implant aftercare (neither in-clinic nor via telehealth). A median additional non-refundable cost for attending clinic appointment was AUD47.50 (range AUD0–700) was noted for our cohort. Thus, cost for using Remote Check™ would need to be absorbed by the CI user, CI clinic, or both. Similar issues relating to the cost of telehealth provision, particularly in Australia, have been observed in other medical fields [45]. A videoconference or call incorporated into the service model (e.g., to discuss results) could help recoup some costs, and was acceptable with survey participants for most situations.

Although participants showed a strong preference against paying an annual fee of AU$120 versus the reference level of AU$10 per test, it is unlikely that clinics would be able to support a remote service with a fee of AU$10 per test. These findings are similar to several other studies which have shown that while people may sometimes be willing to pay for telehealth services, the amount they are prepared to pay, particularly for health apps, is generally very low [19,46,47]. Indeed, a qualitative study of user perceptions of mobile health apps found that 77% of participants only used free apps, and cost was a determining factor in adoption of health apps across all ages and socioeconomic groups [46]. In the current survey, the option of an annual fee (of AU$40) was preferred over a per-test fee of AU$30, which suggests an annual subscription-based service may provide an acceptable and viable payment infrastructure. Furthermore, if a clinic is to consider charging CI users for the cost of remote services, it is vitally important that the value, reliability, security and benefit of using the remote service, used instead of or in combination with an in-clinic service, is clearly demonstrated to potential users.

The importance of perceived value, reliability and benefit for CI clinics and users was highlighted by the cochlear implant home care (CHOICE) program in the UK [13,25]. Independent evaluation of an expanded CHOICE program implementation, carried out by the Wessex Academic Health Science Network, reported a substantially smaller adoption and spread than planned (240 CI users, compared with a target of 2,200) [18,25]. In addition to the impact of the COVID-19 pandemic on the study, the scalability of CHOICE was thought to be affected by low perceptions of its value to users and staff in practice, problems with the functionality of the technology and lack of integration into clinical pathways at implant centres [25].

All participants in the current study indicated that CI users should be told about the availability of Remote Check™. However, less than half of the participants had heard of Remote Check™ prior to completing the survey, despite most

having used CI(s) compatible with Remote Check™ for ≥12 months and being sufficiently technically literate to engage in this CI research survey via email. This lack of informed discussion suggests that clinicians may be unaware of Remote Check™, unwilling or unable to implement a Remote Check™ option in the clinic, or are perhaps making a decision on behalf of individual clients that Remote Check™ would be unsuitable for them. Shared decision making is widely accepted as being essential in the field of audiology, particularly as such decisions can involve complex choices between evidence-based options. Clinicians must provide accurate and understandable information about the benefits and risks of treatment options, and ensure that decisions are sensitive to a person's values and preferences [48–50]. Nevertheless, in a survey about current and future CI service delivery conducted with CI users/carers and professionals in the UK, CI users reported that decisions were currently mainly driven by the implant team [12], which was a finding supported by the focus group discussion and survey in the current study.

Shared decision making must also be considered pre-operatively. Since this DCE was completed, two other major CI manufacturers, Med-El and Advanced Bionics, are soon to or have already released a remote care solution. The specifics of the remote service offered by each manufacturer, rather than simply access to remote services, as presented in this study, may be more relevant to the decision-making process prior to implant in the future.

Although most participants in this study found it easy to make time to attend clinic appointments and the median travel time was reasonable (1 hour), most participants would consider using Remote Check™ for troubleshooting, long-term monitoring, and to complete the speech and hearing component of an appointment. Geographical distance and travel time are often cited as potential barriers for hearing healthcare and as reasons to develop telehealth options [13,14,51]. This study suggests that even those CI users who find it relatively easy to attend clinic appointments are interested in the option of telehealth. This is consistent with a previous Australian study, in which willingness to consider hearing-related telehealth appointments was associated with metropolitan location, younger age and female gender [19]. Our finding that older adults were significantly less likely to accept all types of long-term monitoring remote check appointments except long-term monitoring appointments, and were more likely to prefer in-clinic appointments only is consistent with other studies [19,23] may also be consistent with lower levels of digital literacy amongst older adults noted in Kruse et al's review [2]. Unfortunately we did not assess digital literacy levels in our current study to determine if this was the case for our cohort specifically. Additional barriers to attending in-clinic appointments, such as reluctance to drive in city traffic/ parking issues, poor public transport, and difficulty taking time off work, may drive a preference for telehealth options in urban areas.

It is important to note that a small proportion (12%) of survey participants stated they would not be comfortable utilising Remote Check™ in any of the proposed situations. A hybrid model of care would therefore need to operate, offering remote CI assessments alongside standard, in-clinic follow-up care for different CI users [14]. This finding is echoed by Carner [17] who concluded that it was mandatory to accurately select appropriate CI users for use of Remote Check™ at home [17]. As per the current study, Carner [17] noted the importance of equal digital technology access, privacy, security, and regulation of service payment for all Remote Check™ users when considering implementation of Remote Check™ into regular clinical services in Italy, and national health laws would need to be enacted to ensure this.

## Limitations of the current study

The relatively low completion rate for the survey (32%) means that the responses may not accurately represent the CI user population as a whole. The high drop-out rate for CI users who started the survey (50%) may have been due to the length of the survey, poor comprehension about the purpose of survey, which can be cognitively challenging for some people [52], or frustration regarding the repetitive, and/or confusing nature of the DCE option pairs. This frustration was suggested by some of the participant via comments, such as "I found that if I could pick and choose, in some cases I would have half the elements from Option A and half from Option B" or "I wanted to answer each question separately – sometimes A, sometimes B, but could not. This is a Survey Design 101 issue – one question one answer". These

comments imply that, despite the frustration generated, the DCE met its objective by forcing participants to make a trade-off between attribute levels they preferred and those of less importance to them. Providing instructions and examples via videoconferencing or face-to-face meetings, rather than via email only, may have improved comprehension and partici-pation in the study, but would have been time and resource intensive. The use of email to distribute the survey may have also led to self-selection of CI users possessing at least some degree of digital literacy, thus biasing the responses to some extent.

As CI users recruited for the study were individuals who had previously agreed to be contacted for research commu-nications, it is possible that recruitment method may have biased results positively towards use of remote technology as people interested in research could be more interested in new CI technology. Furthermore, it may be considered that the older age of adult CI users in general, as well as the age of our respondents (median age = 64 years) could negatively bias both completion of, and responses to the survey itself. In fact older respondents were generally less likely to accept future remote check review. Despite this, we note that the median age of our respondents is similar to the average of adult CI users across various large adult CI centres [53] thus representative of the average age of adult CI users in general. The use of digital technology, including remote technology, is becoming increasingly prevalent amongst older adults. Given this, it cannot be assumed that age alone predetermines it's use, but rather that it is also influenced by digital liter-acy and personal circumstance.

The number of participants was too small to examine hypotheses within subgroups with accuracy. For example, it was not possible to assess preferences for participants who found it difficult to find time to attend clinic appointments and/or had long journey times, willingness to pay within different groups (e.g., younger vs older, lower vs higher annual income), nor the effects of age, digital literacy or familiarity with Remote Check on preferences.

### Next steps

Further exploration of the preferences of a more diverse range of CI users is needed to complete our understanding of the current needs. Finding ways to increase the representation of CI users with more difficulty accessing current in-clinic ser-vices, for example due to work or carer responsibilities, financial concerns and/or geographical constraints, and inclusion of more new/prospective CI users would ensure research is informed by a diverse range of perspectives and experience. In addition, with new remote service options available from other CI manufacturers, and expansion of the Cochlear Ltd service, exploration of the preferences of CI users with regard to these offerings is also needed.

It is essential to consider the potential barriers and facilitators for clinic staff when designing a remote care service. It would be valuable to determine provider perspectives on using remote services with a diverse range of clients (e.g., elderly or complex clients) and on the practicality of introducing remote care into the clinic workflow. Technical/adminis-trative barriers associated with implementing a remote service may include: differences in the test battery offered with Remote Check™ versus in-person clinic evaluations; data capture and storage; absence of in-app payment options; and the requirement for different workflow patterns for CI users using Cochlear Ltd devices compatible with Remote Check™ compared with CI users whose devices are not compatible with Remote Check™, or who are unable/unwilling to use remote follow-up services.

### Conclusion

Integration of Remote Check™ into routine clinical care has the potential to streamline delivery of care for CI users, but the conditions required for implementation of remote technologies into clinic systems need to be fully understood for an intervention to be taken up and maintained. Outcomes from this DCE provide valuable insights about the preferences of CI users regarding use of Remote Check™ and may help to inform customisation of remote services by individual audi-ology clinics. For example, clinics may need to consider prioritising clinician continuity, whereas factors such as timing and delivery of results and type of report may be more amenable to customisation to suit the clinic's specific clinical,

organisational and environmental contexts. Costs and payment infrastructure for providing remote care requires careful consideration to ensure costs to the clinic are covered while costs to the user are minimised.

Remote care options such as Remote Check™ should be discussed with all CI candidates, although this is was not the case at the time of this study. For Remote Check™-compatible CI users, timing of the discussion may occur at any time (with a preference for before implant or 3–6 months after implant), but if a choice of implant brand is offered, the availability of remote follow-up options should be discussed before implant, as it could affect brand choice.

Although most participants would consider using Remote Check™ alongside clinic visits (e.g., if a problem was identified) for a wide range of situations, some participants had a preference to attend all appointments in the clinic. A hybrid service model with the option of both Remote Check and in-clinic appointments would therefore be required.

## Supporting information

**S1 Fig. Thematic analysis of focus group discussions with adult CI users (combined outcomes from one group with Remote Check experience [n = 5] and one group with no Remote Check experience [n = 5]).**
(DOCX)

**S1 Table. Statistical analysis of demographic factors associated with willingness to accept a future remote check appointment in various formats.**
(DOCX)

**S2 Table. DCE results – pooled and per scenario (Conditional fixed-effects logistic regression).**
(DOCX)

**S3 Table. DCE design.** Choice pairs were identified using Ngene software. Scenario 1= troubleshooting, Scenario 2=long-term review appointment, Scenario 3=acute care. Attribute levels are listed in table 1 of the main manuscript. For example, Choice set 1 presents a choice pair of the following: OPTION A: reviewed by regular audiologist, provided with a detailed comparison of hearing and speech tests with previous results, feedback provided within a week, information received via a notification in the Nucleus Smart app, at a cost of $120 annual fee for unlimited number of checks. OPTION B: reviewed by a trained admin staff, with a response only if the test indicated a problem, with information provided by the next day, via a face-to-face meeting with the audiologist, at an annual fee of $40 for unlimited checks.
(DOCX)

**S4 Table. DCE block allocation for all participants who started the DCE survey.** Participants with ≥95% survey progress were included in data analysis.
(DOCX)

**S1 File. DCE Survey: Example survey questionnaire showing information provided on Remote Check, one block of discrete choice experiment options, and additional questions on Remote Check.**
(DOCX)

## Acknowledgements

The authors wish to acknowledge the help and support of all individuals and groups involved in the completion of this study. Denise Howting provided assistance with data collection. Medical writing assistance was provided by Dr Susan Cripps of Sue Cripps Medical Communications. The Ear Science Institute Cochlear Implant Clinic, the South Australian Cochlear Implant Clinic, and Cochlear Care Centre (Melbourne) who assisted with survey distribution. And finally, all the cochlear implant users who kindly donated their time to complete the survey, allowing this research to occur.

## Author contributions

**Conceptualization:** Catherine Sucher, Richard Norman, Rebecca Bennett, Melanie Ferguson.

**Data curation:** Catherine Sucher, Emma Chaffey, Rebecca Bennett.

**Formal analysis:** Richard Norman, Emma Chaffey, Rebecca Bennett, Melanie Ferguson.

**Funding acquisition:** Catherine Sucher, Melanie Ferguson.

**Investigation:** Catherine Sucher, Melanie Ferguson.

**Methodology:** Catherine Sucher, Richard Norman.

**Project administration:** Catherine Sucher.

**Software:** Richard Norman.

**Writing – original draft:** Catherine Sucher.

**Writing – review & editing:** Catherine Sucher, Richard Norman, Emma Chaffey, Rebecca Bennett, Melanie Ferguson.

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
