## [Decision Letter · Decision Letter 0]

2 Jan 2025

PONE-D-24-44753Patient preferences for remote cochlear implant management: A discrete choice experimentPLOS ONE

Dear Dr. Sucher,

Thank you for submitting your manuscript to PLOS ONE. After careful consideration, we feel that it has merit but does not fully meet PLOS ONE’s publication criteria as it currently stands. Therefore, we invite you to submit a revised version of the manuscript that addresses the points raised during the review process.

We look forward to receiving your revised manuscript.

Kind regards,

Jorge Spratley, MD, PhD

Academic Editor

PLOS ONE

Additional Editor Comments:

Dear Authors,

In the quality of academic-editor of PlosOne, I could read with interest your manuscript entitled “Patient preferences for remote cochlear implant management: A discrete choice experiment”.

It is a well written, somewhat lengthy article, that addresses a cutting-edge topic on CIs thematic. Its review raised the following comments remarks:

1. It is a pity that it took so long to have the paper submitted since the survey finalization, back in November 2022. Much of its novelty has been lost since then, as some more recent publications have focused on the subject.

2. In Introduction, on the interest of transparency, please state that other CI brands also have similar remote care programs to assist patients with CIs. It is already mentioned en-passant on Discussion, but should also be stated in Introduction.

3. In Material and Methods, it is neither stated the mean age of respondents nor if age dispersion might have influenced the type of responses.

4. Also, it was not clear which was the distribution of time/years since cochlear implantation of respondents and how many were uni- versus bilateral CI users. Did these demographic measures influence the trends of the responses or rate of dropouts (>50%)?

5. Did the authors gather data about the overall satisfaction of the respondents with their own CIs and cross this information with the interest shown to adhere to a remote check of their devices?

6. In page 16, line 274, please remove the whole sentence or, at least, carefully rewrite it. It is overtly commercial biased and therefore unacceptable. Your paper, despite all the appropriate disclosures of interest manifested, is focused on one single commercial brand (Cochlear). This fact demands a very strict compliance to exemption and avoidance of commercial allusions.

7. In page 16, line 266, annum should be in italic.

I also take this opportunity to kindly advise the authors to fully respond to the comments/remarks from Reviewer #1 and to the Editor’s review, highlighting in the revised manuscript all the additions and text changes.

Reviewers' comments:

Reviewer's Responses to Questions

**Comments to the Author**

1. Is the manuscript technically sound, and do the data support the conclusions?

Reviewer #1: Yes

2. Has the statistical analysis been performed appropriately and rigorously? 

Reviewer #1: Yes

3. Have the authors made all data underlying the findings in their manuscript fully available?

Reviewer #1: Yes

4. Is the manuscript presented in an intelligible fashion and written in standard English?

Reviewer #1: Yes

5. Review Comments to the Author

Reviewer #1: Overall Qualitative Evaluation

Thank you for submitting your manuscript for evaluation. I have carefully reviewed the text and found it suitable for the journal. The original proposed topic presents a fresh perspective that could help optimise cochlear implant programming and patient adherence to programming sessions. The paper presents an ethical commitment, identifies the founding source, and is adequate in length and organisation.

The manuscript is presented clearly and logically. It complies with the instructions for authors.

I recommend publication after minor reviews. The paper addresses a contemporary cochlear implant issue in a new and original way using a discrete choice experiment.

**Title**

The title is informative and concise, reflecting the study's content and importance. A short title is made available as per the publisher’s instructions.

**Abstract**

It concisely summarises the aims, key methods, important findings, and conclusions. It reflects the manuscript's content, summarizes it, and stands alone.

Although the abbreviations are clearly indicated, their presence goes against the publisher’s rules, so they should be avoided.

Has the correct length.

I think the coma is misplaced on the last line (46): it should be “preferred by, all cochlear implant users.”

**Introduction**

Provide the general context and background of the topic, summarising relevant and current literature.

The study's problem is well addressed and justified, citing all relevant references.

The objective of the study is clearly stated in the last paragraph.

On line 118, it is said, “These tests are not directly comparable to in-clinic audiogram testing [17]. However, in my understanding of the cited paper, it is said, “In all but one participant of this study, the RC application outcomes were the same as in clinic assessment when determining whether the CI users required any further clinical action. Chi-square analysis found that the number of CI users (79/80, 99%) where the test battery was successfully identified all issues recognized by the clinician during the in-presence session, was statistically significant (p < 0.05).” and the same conclusions had two other authors cited by the cited paper. So, please rephrase the sentence to be more accurate.

**Methods**

The design and methodology are original and appropriate to this study.

It is not clear how the participant selection is made. Is it chronological, opportunity, or other criteria? What are the inclusion and exclusion criteria?

The exposure and intervention in which outcomes were measured are well-defined.

The statistical techniques used are well described.

The Ethics Committee's approval for the study is clearly remarked.

In lines 188-9, the “ease of making time to attend CI appointments” is mentioned, but the question does not show how and when it is asked. I suggest it be added to the “supplementary data.”

**Results**

The data presentation is rigorous, clear and convincing.

Tables and figures are legible and properly designed.

In line 213, the sentence “Almost all participants had used a CI at least one year for ≥ 6 months” doesn’t make sense. Please rephrase it.

The only result not present in Table 2 or the supplementary data is referred to in lines 213-15. It could be clearer if it was so contemplated, considering that it is referred to again in line 416 in the Discussion.

**Discussion**

Does explain the implications of the results.

In lines 372-85, the price is discussed without referring to the present in-clinic price. If so, the discussion would be more exact.

In line 81, one study is cited saying that patients over 50 years of age have more barriers when using remote APPs. Could this be the answer to the low adhesion rate in this survey since most of the patients are over 50?

The limitations and areas that need further study are identified. But being this a post-CI survey, couldn’t the experience with the CI and the department be a bias in the participation rate and the answers? Do you think a prospective, theoretical survey, before doing the CI, has different answers and adhesion rates?

**Conclusion**

The conclusions are relevant and related to the objectives.

**References**

The existing literature has been appropriately considered, and the articles follow the correct style.

6. PLOS authors have the option to publish the peer review history of their article (what does this mean? ). If published, this will include your full peer review and any attached files.

**Do you want your identity to be public for this peer review?** For information about this choice, including consent withdrawal, please see our Privacy Policy .

Reviewer #1: No

---

## [Author Response · Author response to Decision Letter 1]

9 Feb 2025

Dear Dr Sprately,

Thank you for giving us the opportunity to submit a revised draft of the manuscript “Patient preferences for remote cochlear implant management: A discrete choice experiment” (manuscript no.: PONE-D-24-44753) for publication in PLOS ONE. We appreciate the time and effort that you and the reviewers dedicated to providing feedback on our manuscript and are grateful for the insightful comments on and valuable improvements to our paper. We have incorporated most of the suggestions made by the reviewers. Those changes are highlighted within the manuscript. Please see below, in blue, for a point-by-point response to the reviewers’ comments and concerns. All line numbers refer to the revised manuscript file with tracked changes.

Yours Sincerely,

Cathy Sucher, AuD

Catherine.sucher@uwa.edu.au

Adjunct Research Fellow, School of Medicine

The University of Western Australia.

Author Response to Review comments

Additional Editor Comments

In the quality of academic-editor of PlosOne, I could read with interest your manuscript entitled “Patient preferences for remote cochlear implant management: A discrete choice experiment”. It is a well written, somewhat lengthy article, that addresses a cutting-edge topic on CIs thematic.

Author response: Thank you very much

1. It is a pity that it took so long to have the paper submitted since the survey finalization, back in November 2022. Much of its novelty has been lost since then, as some more recent publications have focused on the subject.

Author response: Yes, we agree that unfortunately it did take some time to submit this research. However, we believe that the research remains novel in that no new research presents patient preferences for implementation of remote services, or utilises a Discrete choice experiment to do so.

2. In Introduction, on the interest of transparency, please state that other CI brands also have similar remote care programs to assist patients with CIs. It is already mentioned en-passant on Discussion, but should also be stated in Introduction.

Author response: Thank you for your suggestion. We have added the following to the introduction (lines 77-80):

“Several published studies have examined the feasibility of remote care options for CI users, and have highlighted the desirability of remote care [14-22] both using manufacturer-based and stand-alone remote care options. Furthermore, since completion of this study, all three major CI manufacturers, Cochlear Ltd, Med-El and Advanced Bionics have released remote care options.”

The following statement has also been added to the next steps section of the manuscript (lines 528-531);

“In addition, with new remote service options available from other CI manufacturers, and expansion of the Cochlear Ltd service, exploration of the preferences of CI users with regard to these offerings is also needed.”

3. In Material and Methods, it is neither stated the mean age of respondents nor if age dispersion might have influenced the type of responses.

Author response: We believe that the digital literacy and personal circumstances, rather than the age alone of respondents may have influenced responses given to the DCE as well as biased completion of the survey itself. Use of digital technology, including remote technology, is becoming increasingly prevalent amongst older adults, thus we do not wish to make the assumption that age alone predetermines it’s use. Furthermore, the average age of our respondents is consistent with the average age of adult CI users across a number of countries (Goudey et al, 2021) thus we feel that the data set is representative of adult CI users at least in age. Having said that, we have performed additional analysis that indicates that older CI recipients are significantly less likely to consider accepting future remote check review appointments for all types of appointment format except long-term monitoring. This analysis is shown in Supplementary table 1. The following as been added to the manuscript at lines 213-217

“Demographic factors were analysed to determine which factors were associated with individuals’ willingness to consider remote check reviews of various formats in the future. A combination of t-tests (where data were normally distributed, ranked sign tests (where categorical data was recorded and could be ranked in order of strength) and Chi2 tests for categorical data that could not be ranked, were used for analysis.”

Line 233

“The median age of participants was 64 years (range 26-89, mean (SD)=63.7±13.1).”

Lines 249-265

“Participants were asked to select one or more options relating to the type of remote check review they may consider using in the future. These included, 1) trouble-shooting appointment, 2) additional monitoring during the first 6 months post switch-on, 3) completion of the speech and hearing component of their annual review, 4) a standalone annual review without coming into clinic, 5) only coming into the clinic, no remote check. Only 15 (12%) of participants would not consider any form of remote check review appointment in the future. Statistical analysis of factors which could influence individuals’ consideration of remote check reviews is shown in Supplementary table 1.”

Older participants were significantly less likely to accept remote check appointments for trouble-shooting (t=-2.6, df=121, p=0.012), additional monitoring in the first 6 months post-CI (U(46,77) = 1190.5. p=0.002), completion of speech and hearing assessment (t=-3.2, DF=121. P=0.002), and more likely to only want in-clinic appointments only (t=-2.7, DF=122. P=0.007). Participants with lower annual household incomes were significantly more likely to reject troubleshooting appointments (U(81,42) = 676.5. p=0.035), and participants less able to make time for in-clinic appointments were significantly more likely to accept long-term monitoring remote check appointments (U(81,42) = 1258.0. p=0.042. No other demographic factors were significantly related to consideration of future remote checks”

Lines 473-478

“Our finding that older adults were significantly less likely to accept all types of long-term monitoring remote check appointments except long-term monitoring appointments, and were more likely to prefer in-clinic appointments only is consistent with other studies [19, 23] may also be consistent with lower levels of digital literacy amongst older adults noted in Kruse et al’s review [2} Unfortunately we did not assess digital literacy levels in our current study to determine if this was the case for our cohort specifically.”

It was stated in the methodology that all CI users eligible to use remote check ≥18 years of age were invited to complete the survey (line 139). However, to improve clarity, that section has been reworded to the following (lines 145-155).

“Potential participants were recruited opportunistically through contacts at Cochlear Australia, the Ear Sciences Institute Australia (Perth), and the Cochlear Care Centre (East Melbourne) with invitations to participate emailed to 382 adult CI users identified as meeting inclusion criteria; adults aged ≥ 18 years, using Cochlear Nucleus implants (CI632, CI622, CI612, CI532, CI522, CI512, CI24M, CI24R and CI24RE) implant/s and sound processor/s (Nucleus 7 and Kanso 2) compatible with Remote CheckTM (Nucleus 7 or 8) and who had used at least one CI for 6 months or more. All individuals emailed had agreed to be contacted for research communications and had provided a contact email”.

The median age of respondents (64 year) is provided in the results section in table 2 and we felt that this is the more appropriate location for this information (see line 244 of the manuscript).

The following statement has been added to the manuscript in the limitations section (lines 509-521)

“Furthermore, it may be considered that the older age of adult CI users in general, as well as the age of our respondents (median age = 64 years) could negatively bias both completion of, and responses to the survey itself. In fact older respondents were generally less likely to accept future remote check review. Despite this, we note that the median age of our respondents is similar to the average of adult CI users across various large adult CI centres [52] thus representative of the average age of adult CI users in general. The use of digital technology, including remote technology, is becoming increasingly prevalent amongst older adults. Given this, it cannot be assumed that age alone predetermines it’s use, but rather that it is also influenced by digital literacy and personal circumstance.”

4. Also, it was not clear which was the distribution of time/years since cochlear implantation of respondents and how many were uni- versus bilateral CI users. Did these demographic measures influence the trends of the responses or rate of dropouts (>50%)?

Author response: Thank you for your comment. Both duration of use, based on the first ear implanted, and device configuration information are provided in table 2 (line 244 of the manuscript) for individuals who completed the survey.

As the survey was quite lengthy, some of the demographic questions were asked at the beginning of the survey, and some at the end. Duration of use, and device configuration questions were asked at the end of the online survey. There were a number of participants who failed to complete the survey up to this point, thus we are unable to determine if these factors influenced the rate of drop outs.

With regard to duration of use, respondents were asked to identify which of a range of options best represented their duration of device use. Percentages of respondents within each category, rather than simply the number of respondents within each category, have been added to Table 2. In addition, the following statement has been added to the manuscript;

“Ninety-six percent of participants had been using their first implanted CI for a year or more.” (line 233)

With regard to device configuration,

Fitting configuration is also reported in table 2. Additional descriptions of each category have been added to the table to improve clarity.

Additional statistical analyses (Rank Sum tests) have been performed to determine if device configuration or duration of device use influenced willingness to accept a remote check appointment. These results are shown in Supplementary Table 4, which has been added as new data. Statistically significant differences are reported in lines 249-265.

“Participants were asked to select one or more options relating to the type of remote check review they may consider using in the future. These included, 1) trouble-shooting appointment, 2) additional monitoring during the first 6 months post switch-on, 3) completion of the speech and hearing component of their annual review, 4) a standalone annual review without coming into clinic, 5) only coming into the clinic, no remote check. Only 15 (12%) of participants would not consider any form of remote check review appointment in the future. Statistical analysis of factors which could influence individuals’ consideration of remote check reviews is shown in Supplementary table 1. Older participants were significantly less likely to accept remote check appointments for trouble-shooting (t=-2.6, df=121, p=0.012), additional monitoring in the first 6 months post-CI (U(46,77) = 1190.5. p=0.002), completion of speech and hearing assessment (t=-3.2, df=121. p=0.002), and more likely to only want in-clinic appointments only (t=-2.7, df=122. p=0.007). Participants with lower annual household incomes were significantly more likely to reject troubleshooting appointments (U(81,42) = 676.5. p=0.035), and participants less able to make time for in-clinic appointments were significantly more likely to accept long-term monitoring remote check appointments (U(81,42) = 1258.0. p=0.042. No other demographic factors were significantly related to consideration of future remote checks.”

”

5. Did the authors gather data about the overall satisfaction of the respondents with their own CIs and cross this information with the interest shown to adhere to a remote check of their devices?

Author response: Thank you for your observation. The authors did gather information regarding respondents’ overall satisfaction with their own CI’s however this was not reported in the paper. We did not gather information on adherence to remote check assessments, but rather asked respondents about their willingness to receive remote check assessments in the future for particular appointment types, including troubleshooting, post-CI early-stage monitoring, to complete the speech and hearing part of their long-term review, for long term monitoring, or to not receive remote check appointments at all. We have therefore since added the satisfaction information to Table 2 (Line 244). Additional statistical analyses (Rank Sum tests) have been performed to determine if device satisfaction influenced willingness to accept a remote check appointment. These results are shown in Supplementary Table 4, which has been added as new data.

6. In page 16, line 274, please remove the whole sentence or, at least, carefully rewrite it. It is overtly commercial biased and therefore unacceptable. Your paper, despite all the appropriate disclosures of interest manifested, is focused on one single commercial brand (Cochlear). This fact demands a very strict compliance to exemption and avoidance of commercial allusions.

Author response: Unfortunately, it was not possible to ask about other manufacturer’s remote care options at the time of the survey as they were not available. However, we acknowledge that the way the sentence is written could be seen as commercially biased. We have reworded the sentence as below in an attempt to rectify this issue (lines 306-312);

“All survey participants indicated that audiologists should discuss the option of remote care options with CI users at some point in their CI journey (Figure 3). The most commonly proposed timings for such a discussion were ‘When discussing which device to get prior to having the implant surgery’ or ‘After the implant fitting is stabilised and the CI user is comfortable with how the implant works (3–6 months after implantation)’. Over 40% of participants indicated that an awareness of the ability to access their cochlear implant care remotely post-operatively, may have influenced their choice of CI brand.”

7. In page 16, line 266, annum should be in italic.

Author response: Thank you we have made this change to the manuscript.

Reviewers' comments:

Overall Qualitative Evaluation

Thank you for submitting your manuscript for evaluation. I have carefully reviewed the text and found it suitable for the journal. The original proposed topic presents a fresh perspective that could help optimise cochlear implant programming and patient adherence to programming sessions. The paper presents an ethical commitment, identifies the founding source, and is adequate in length and organisation.

The manuscript is presented clearly and logically. It complies with the instructions for authors.

I recommend publication after minor reviews. The paper addresses a contemporary cochlear implant issue in a new and original way using a discrete choice experiment.

Author response: Thank you very much

1. Abstract: It concisely summarises the aims, key methods, important findings, and conclusions. It reflects the manuscript's content, summarizes it, and stands alone.

Although the abbreviations are clearly indicated, their presence goes against the publisher’s rules, so they should be avoided.

Author response: All abbreviations have been removed from the abstract

2. Abstract: I think the coma is misplaced on the last line (46): it should be “preferred by, all cochlear implant users.”

Author response: Thank you. This has been changed as per above comment.

3. Introduction: On line 118, it is said, “These tests are not directly comparable to in-clinic audiogram testing [17]. However, in my understanding of the cited paper, it is said, “In all but one participant of this study, the RC application outcomes were the same as in clinic assessment when determining whether the CI users r

---

## [Editor Report · Decision Letter 1]

19 Feb 2025

Patient preferences for remote cochlear implant management: A discrete choice experiment

PONE-D-24-44753R1

Dear Dr. Sucher,

Thank you for your time and effort to complete the revision of the manuscript in a comprehensive manner. We are pleased to inform you that your manuscript has been judged scientifically suitable for publication and will be formally accepted for publication once it meets all outstanding technical requirements.

Kind regards,

Jorge Spratley, MD, PhD

Academic Editor

PLOS ONE

---

## [Editor Report · Acceptance letter]

PONE-D-24-44753R1

PLOS ONE

Dear Dr. Sucher,

I'm pleased to inform you that your manuscript has been deemed suitable for publication in PLOS ONE. Congratulations! Your manuscript is now being handed over to our production team.

Kind regards,

on behalf of

Professor Jorge Spratley

Academic Editor

PLOS ONE